# Lecanicilliums A–F, Thiodiketopiperazine-Class Alkaloids from a Mangrove Sediment-Derived Fungus *Lecanicillium kalimantanense*

**DOI:** 10.3390/md21110575

**Published:** 2023-10-31

**Authors:** Lin-Fang Zhong, Juan Ling, Lian-Xiang Luo, Chang-Nian Yang, Xiao Liang, Shu-Hua Qi

**Affiliations:** 1CAS Key Laboratory of Tropical Marine Bio-Resources and Ecology, Guangdong Key Laboratory of Marine Materia Medica, South China Sea Institute of Oceanology, Chinese Academy of Sciences, Guangzhou 510301, China; 18851107875@163.com (L.-F.Z.); lingjuan@scsio.ac.cn (J.L.); liangxiao@scsio.ac.cn (X.L.); 2University of Chinese Academy of Sciences, Beijing 100049, China; 3The Marine Biomedical Research Institute, Guangdong Medical University, Zhanjiang 524023, China; luolianxiang321@gdmu.edu.cn (L.-X.L.); nicholasyeung@gdmu.edu.cn (C.-N.Y.)

**Keywords:** mangrove sediment-derived fungus, *Lecanicillium kalimantanense*, thiodiketopiperazine-class alkaloid, cytotoxicity, antibacterial

## Abstract

Six new thiodiketopiperazine-class alkaloids lecanicilliums A–F were isolated from the mangrove sediment-derived fungus *Lecanicillium kalimantanense* SCSIO41702, together with thirteen known analogues. Their structures were determined by spectroscopic analysis. The absolute configurations were determined by quantum chemical calculations. Electronic circular dichroism (ECD) spectra and the structure of Lecanicillium C were further confirmed by a single-crystal X-ray diffraction analysis. Lecanicillium A contained an unprecedented 6/5/6/5/7/6 cyclic system with a spirocyclic center at C-2′. Biologically, lecanicillium E, emethacin B, and versicolor A displayed significant cytotoxicity against human lung adenocarcinoma cell line H1975, with IC_50_ values of 7.2~16.9 μM, and lecanicillium E also showed antibacterial activity against four pathogens with MIC values of 10~40 μg/mL. Their structure–activity relationship is also discussed.

## 1. Introduction

Natural products containing sulfur exhibit significant structural diversities and diverse biological activities [1,2]. Among them, thiodiketopiperazine, as an interesting subgroup, is an important class of secondary metabolites of fungi, especially species of *Aspergillus fumigatus*, *A. terreus*, *A. flavus*, *A. niger*, *Trichoderma virens*, *T. viride*, and *Penicillium terlikowskii* [3]. The structures of thiodiketopiperazines are diverse [4,5], of which the spirocyclic thiodiketopiperazines are rare in nature [5,6]. Many thiodiketopiperazines have been reported to display a wide range of biological properties, including antiproliferative, cytotoxic, antibacterial, antifungal, antiviral, and anti-angiogenic activities [7,8]. For examples, brocazine G displayed potent antibacterial activity against *Staphylococcus aureus* with a MIC value of 0.25 μg/mL and strong cytotoxicity against human ovarian cancer cells (IC_50_ = 0.66 μM) [8]. Penicisulfuranols A–C showed cytotoxicity towards human cervical carcinoma cell lines and human promyelocytic leukemia cells with IC_50_ values of 0.1–3.9 μM [9]. Penicibrocazine C exhibited antibacterial activity against *Micrococcus luteus* with MIC value of 0.25 μg/mL, which was stronger than the positive control, chloromycetin (MIC = 2.0 μg/mL) [10]. Numerous structurally unique and potential variety activities of the thiodiketopiperazines have drawn much attention from synthetic chemists and pharmacologists.

The fungal genus *Lecanicillium* (formerly *Verticillium*), including *L. fusisporum*, *L. psalliotae,* and *L. lecanii,* are pathogenic fungi that are widely used as biological pesticides [11,12,13]. The secondary metabolisms of these fungi are rarely reported [11]. Only a few compounds belonging to indolosesquiterpenoids, phenopicolinic acid derivatives, tetracyclic diterpenoids, pregnanes, and cyclic lipodepsipeptides have been isolated from the genus of *Lecanicillium* up to now [14,15,16]. The fungus *L. kalimantanense* SCSIO41702 was isolated from a mangrove sediment sample collected from Hainan province. There was no report about the secondary metabolisms of *L. kalimantanense*. In order to explore novel, natural compounds from fungi, we investigated the secondary metabolisms of *L. kalimantanense* SCSIO41702, which led to the isolation of six new thiodiketopiperazine-class alkaloids, lecanicilliums A-F (**1**–**6**), together with thirteen known analogues (Figure 1): emethacin B (**7**) [17], bisdethiobis(methylsulfanyl)acetylaranotin (**8**) [8], bisdethiobis(methylsulfanyl)deacetylaranotin (**9**) [8], bisdethiobis(methylsulfanyl)-aranotin (**10**) [8], 8-acetyl-bisdethiobis(methylsulfanyl)apoaranotin (**11**) [18], bisdethiobis(methylsulfanyl)acetylapoaranotin (**12**) [8], bisdethiobis(methylsulfanyl)apoaranotin (**13**) [19], bisdethiobis(methylsulfanyl)deacetylapoaranotin (**14**) [8], haematocin (**15**) [8], versicolor A (**16**) [20], 12,12a-anhydro-desacetyl-bis-dethio-7a, 14a-di-(methylmercapto)-apoaranotin (**17**) [21], emestrin H (**18**) [22], and citriperazine B (**19**) [23]. These compounds were evaluated for their cytotoxicity, toxicity against brine shrimps, and antibacterial activity. Herein, we report the isolation and structural elucidation as well as the biological activities of these compounds.

## 2. Results and Discussion

Lecanicillium A (**1**) was obtained as a white powder with the molecular formula C_18_H_16_N_2_O_5_S as determined by HRESIMS ([M + H]^+^
*m*/*z* 373.0851) and NMR data. The ^1^H NMR spectrum (Table 1) displayed the presence of two active hydrogens at *δ*_H_ 10.06 (^1^H, s), 5.31 (^1^H, s), four aromatic hydrogens at *δ*_H_ 7.26 (^1^H, d, *J* = 7.5 Hz), 7.14 (^1^H, t, *J* = 8.0 Hz), 6.92 (^1^H, t, *J* = 7.5 Hz), 6.81 (^1^H, d, *J* = 8.0 Hz), two olefinic methines at *δ*_H_ 6.00 (^1^H, d, *J* = 6.5 Hz), 4.82 (^1^H, ddd, *J* = 7.8, 6.5, 1.2 Hz), four tertiary methine groups at *δ*_H_ 6.61 (^1^H, d, *J* = 6.5 Hz), 4.86 (^1^H, ddd, *J* = 7.8, 6.5, 1.2 Hz), 3.87 (^1^H, d, *J* = 6.5 Hz), 3.06 (^1^H, t, *J* = 6.5 Hz), and two methylenes at *δ*_H_ 3.94 (^1^H, d, *J* = 16.4 Hz), 3.26 (^1^H, d, *J* = 16.4 Hz), 2.71 (^1^H, dd, *J* = 12.7, 6.5 Hz), 2.21 (^1^H, d, *J* = 12.7 Hz). The ^13^C NMR spectrum (Table 2) displayed 18 carbon signals including two methylenes, ten methines (six aromatic/olefinic and three oxygenated or heteroatomic), and six nonprotonated carbons (two diagnostic amide carbonyl, two olenfinic and two heteroatomic). These NMR data of **1** (Table 1 and Table 2) showed similarity to those of spirobrocazines A–C [5], which suggested that **1** was a thiodiketopiperazine alkaloid. Detailed analysis of HSQC, HMBC and COSY spectra (Figure 2) proved that **1** contained the same structural parts of rings A–C as those of spirobrocazines A–C. However, the COSY spectrum (Figure 2) showing correlations from H-4 (*δ*_H_ 3.06, ^1^H, t, *J* = 6.5 Hz) to H_b_-3 [*δ*_H_ 2.71 (^1^H, dd, *J* = 12.7, 6.5 Hz)], H-5 (*δ*_H_ 3.87, ^1^H, d, *J* = 6.5 Hz) and H-9 (*δ*_H_ 6.61, ^1^H, d, *J* = 6.5 Hz), from H-5 to H-6 (*δ*_H_ 4.86, ^1^H, ddd, *J* = 7.8, 6.5, 1.2 Hz), and from H-7 (*δ*_H_ 4.82, ^1^H, ddd, *J* = 7.8, 6.5, 1.2 Hz) to H-6 and H-8 (*δ*_H_ 6.00, ^1^H, d, *J* = 6.5 Hz), and the HMBC spectrum (Figure 2) showing correlations from H-4 to C-2 (*δ*_C_ 70.8), C-3 (*δ*_C_ 48.2), C-6 (*δ*_C_ 52.5), from OH-5 to C-5 (*δ*_C_ 69.9), from H-6 to C-2, C-4 (*δ*_C_ 46.3), C-5, C-7, C-8 (*δ*_C_ 144.9), from H-8 (*δ*_H_ 6.00, ^1^H, d, *J* = 6.5 Hz) to C-6, C-7 (*δ*_C_ 100.6), C-9 (*δ*_C_ 92.9), from H-9 to C-2, C-5, C-8, and from H-3 to C-2, C-4, C-9, suggested the presence and assignment of D/E rings in **1** as shown. Combining the molecular formula and the chemical shifts of C-2 and C-6/H-6, it was reasonable to infer a sulfide bond between C-2 and C-6 to form a sulfide six-membered-ring. Thus, the planar structure of **1** was determined as shown.

The relative configuration of **1** was elucidated by analysis of the NOESY spectrum (Appendix A). The NOE correlations of H-4 with OH-5 and H-9, H-9 with H-3 (*δ*_H_ 2.71) and H-4, and H-6 with H-4 (Figure 3), indicated their cofacial β-orientations. The NOE correlation of H-5 with H_a_-3 (*δ*_H_ 2.21) indicated their cofacial *α*-orientations. The NOE correlations from H-5 to H_2_-3′ revealed their cofacial orientations and the CH_2_-3′ group at the axial position. The coupling constant of *J*_H-7-H-8_ = 6.5 Hz indicated that the double bond was a *Z*-configuration. The absolute configuration of **1** was further determined by electronic circular dichroism (ECD) calculations. The calculated ECD spectrum of (2*R*, 4*S*, 5*S*, 6*S*, 9*R*, 2′*S*)**-1** showed similarity to the experimental ECD spectrum of **1** (Figure 4), which confirmed that the absolute configuration of **1** was 2*R*, 4*S*, 5*S*, 6*S*, 9*R*, 2′*S*.

A possible biosynthetic pathway of **1** is shown in Appendix A [5,8]. The biosynthetic pathway of **1** likely starts with the cyclodipeptide (I) composed of two phenylalanines followed by oxidation to afford the key intermediate II, which could be transferred to III by dehydration and oxidation. Then cyclization and oxidation of III via intermediate IV could produce V. Finally, V could be translated into compound **1** by further oxidation, cyclization, and sulfurization.

Lecanicillium B (**2**) was obtained as a white powder with the molecular formula C_19_H_18_N_2_O_5_S as determined by HRESIMS ([M + H]^+^
*m*/*z* 387.1009) and NMR data. The ^1^H and ^13^C NMR spectra of **2** (Table 1 and Table 2) showed similarity to those of spirobrocazine A [5]. The obvious difference between them was the chemical shift changes of C-4, C-5, C-6, C-7, C-8, and C-9 (*δ*_C_ 134.1 (C), 120.3 (CH), 124.0 (CH), 131.6 (CH), 75.1 (CH), and 70.4 (CH) in spirobrocazine A [5], and correspondingly *δ*_C_ 109.1 (C), 136.7 (CH), 137.9 (CH), 111.0 (CH), 71.2 (CH) and 64.1 (CH) in **2**, respectively). The COSY spectrum (Figure 2) showing the sequential correlations of H-6/H-7/H-8/H-9, and the HMBC spectrum (Figure 2) showing correlations from H-5 (*δ*_H_ 6.69, ^1^H, t, *J* = 2.4 Hz) to C-3 (*δ*_C_ 40.2), C-4 (*δ*_C_ 109.1), C-6 (*δ*_C_ 137.9), C-9 (*δ*_C_ 64.1), from H-6 (*δ*_H_ 6.29, ^1^H, dd, *J* = 8.2, 2.4 Hz) to C-5 (*δ*_C_ 136.7), C-7 (*δ*_C_ 111.0), C-8 (*δ*_C_ 71.2), from H-8 (*δ*_H_ 4.46, ^1^H, ddt, *J* = 7.7, 6.7, 2.1 Hz) to C-7, C-9, from OH-8 (*δ*_H_ 5.28, d, *J* = 6.7 Hz) to C-8, and from H-3 [*δ*_H_ 2.91 (^1^H, dt, *J* = 15.3, 1.2 Hz), 3.14 (^1^H, dt, *J* = 15.3, 2.2 Hz)] to C-2 (*δ*_C_ 69.2), C-4, C-5, C-9, suggested that the E ring in **2** was a seven-membered 4,5-dihydrooxepine ring, as shown, instead of a six-membered ring. The coupling constant of *J*_H-6-H-7_ = 8.2 Hz indicated that the double bond was a *Z*-configuration. In addition, the NOESY sepctrum of **2** (Appendix A) also showed great similarity to that of spirobrocazine A. The NOE correlation of OH-8 with H-9 indicated their cofacial β-orientations, while the NOE correlations of H-8 with H_3_-10 and H_a_-3′ (*δ*_H_ 4.01) indicated their cofacial *α*-orientations and the CH_2_-3′ group at the axial position (Figure 3). Furthermore, the calculated ECD spectrum of (2*R*, 8*S*, 9*S*, 2′*S*)**-2** showed similarity to the experimental ECD spectrum of **2** (Figure 4), indicating the absolute configuration of **2,** as shown. 

Lecanicillium C (**3**) was isolated as a white powder with the molecular formula C_24_H_26_N_2_O_9_S_2_ as determined by HRESIMS ([M + Na] ^+^
*m*/*z* 573.0969) and NMR. The ^1^H and ^13^C NMR spectra of **3** (Table 1 and Table 2) showed similarity to those of bisdethiobis(methylsulfanyl)acetylapoaranotin [8]. The obvious difference between them was the additional presence of three tertiary methine signals (*δ*_H_ 3.83 (^1^H, d, *J* = 2.7 Hz), 3.67 (^1^H, dd, *J* = 4.3, 2.7 Hz), 3.34 (^1^H, dd, *J* = 4.3, 1.2 Hz); *δ*_C_ 54.1, 53.6, 47.9) and one oxygenated nonprotonated carbon (*δ*_C_ 60.9), and the disappearance of the signals for two double bonds in **3**. Detailed analysis of HSQC, HMBC and COSY spectra (Figure 2, Appendix A) proved that **3** contained the same structural parts of rings B–E as those of bisdethiobis(methylsulfanyl)acetylapoaranotin. Furthermore, the COSY spectrum (Figure 2) showed correlations from H-5′ (*δ*_H_ 3.83, ^1^H, d, *J* = 2.7 Hz) to H-6′ (*δ*_H_ 3.67, ^1^H, dd, *J* = 4.3, 2.7 Hz), from H-6′ to H-7′ (*δ*_H_ 3.34, ^1^H, dd, *J* = 4.3, 1.2 Hz), and from H-8′ (*δ*_H_ 5.79, ^1^H, dd, *J* = 9.1, 1.2 Hz) to H-7′ and H-9′ (*δ*_H_ 4.34, ^1^H, d, *J* = 9.1 Hz), and the HMBC spectrum (Figure 2) showed correlations from H-5′ to C-4′ (*δ*_C_ 60.9) and C-6′ (*δ*_C_ 47.9), from H-6′ to C-4′, C-5′ (*δ*_C_ 53.6), from H-7′ to C-8′ (*δ*_C_ 72.6), C-9′ (*δ*_C_ 57.5), from H-8′ to C-9′, C-10′ (*δ*_C_ 169.4), from H-9′ to C-1 (*δ*_C_ 163.5), C-4′, C-5′, C-8′, from H-3′ [*δ*_H_ 2.85 (^1^H, d, *J* = 14.4 Hz), 2.43 (^1^H, d, *J* = 14.4 Hz)] to C-1′ (*δ*_C_ 163.6), C-2′ (*δ*_C_ 71.7), C-4′, C-5′, and from H_3_-11′ to C-10′. Combining with the molecular formula, these above data suggested that the A ring was a six-membered ring with an acetyl group attached at C-8′ and two oxygenated three-membered rings between C-4′ and C-5′, and between C-6′ and C-7′, respectively. The small coupling constants of *J*_H-5′-H-6′_ (2.7 Hz), *J*_H-6′-H-7′_ (4.3 Hz), and *J*_H-7′-H-8′_ (1.2 Hz) suggested a *cis*-diaxial relationship between H-5′ and H-6′, between H-6′ and H-7′, and between H-7′ and H-8′, respectively. NOE correlations of H-5′ with H-6′, H-6′ with H-7′ and H-7′ with H-8′ also indicated H-5′, H-6′, H-7′, and H-8′ on the same face. The large coupling constants of *J*_H-8-H-9_ (8.1 Hz) and *J*_H-8′-H-9′_ (9.1 Hz) suggested a *trans*-diaxial relationship between H-8 and H-9, and between H-8′ and H-9′, respectively. Furthermore, the NOE correlations of H-3 (*δ*_H_ 3.25) with H-8 and H_3_-12 indicated the assignment of *α*-orientation for H-8 and H_3_-12, NOE correlations of H-3′ (*δ*_H_ 2.43) with H-5′ and H_3_-12′ indicated *α*-orientation for H_3_-12′ and H-5′, and NOE correlations of H-9 with H-9′ indicated β-orientation for H-9 and H-9′ (Figure 3). Thus, the structure of **3** was inferred as shown. The absolute configuration of **3** was determined as 2*R*, 8*S*, 9*S*, 2′*R*, 4′*S*, 5′*R*, 6′*R*, 7′*R*, 8′*R*, 9′*R* by a single crystal X-ray diffraction analysis using Cu K*α* radiation (Figure 5), which was supported by the calculated ECD spectrum of (2*R*, 8*S*, 9*S*, 2′*R*, 4′*S*, 5′*R*, 6′*R*, 7′*R*, 8′*R*, 9′*R*)-**3** showing agreement with the experimental ECD spectrum of **3** (Figure 4).

Lecanicillium D (**4**) was isolated as a white solid with the molecular formula C_24_H_26_N_2_O_9_S_2_ as determined by HRESIMS ([M + NH_4_]^+^
*m*/*z* 568.1408). The ^1^H and ^13^C NMR spectra of **4** (Table 1 and Table 2) showed similarity to those of **3**. The obvious difference between them was the presence of an *α*, β-unsaturated ketone group (*δ*_H_ 6.98 (^1^H, d, *J* = 10.3 Hz), 6.09 (^1^H, d, *J* = 10.3 Hz); *δ*_C_ 191.5, 150.2, 125.8) and the disappearance of three tertiary methine signals in **4**. The HMBC spectrum (Figure 2) showing correlations from H-5′ (*δ*_H_ 6.98, ^1^H, d, *J* = 10.3 Hz) to C-7′ (*δ*_C_ 191.5) and C-9′ (*δ*_C_ 69.1), from H-6′ (*δ*_H_ 6.09, ^1^H, d, *J* = 10.3 Hz) to C-8′ (*δ*_C_ 74.8) and C-4′ (*δ*_C_ 75.5), from H-8′ (*δ*_H_ 5.73, ^1^H, d, *J* = 11.2 Hz) to C-7′ and C-10′, from H-3′ [*δ*_H_ 2.94 (^1^H, d, *J* = 15.4 Hz), 3.16 (^1^H, dd, *J* = 15.4, 2.1 Hz)] to C-4′ and C-5′, and from H_3_-11′ to C-10′, and the COSY spectrum (Figure 2) showing correlations from H-5′ to H-6′, and from H-8′ to H-9′, revealed the existence of an *α*, β-unsaturated ketone instead of two oxygenated three-membered rings in **4**. The NOE correlation of H-9 with H_3_-11 suggested the assignment of *α*-oriention for H-8 and β-oriention for H-9, and the NOE correlations of H-3 (*δ*_H_ 2.92) with H-9, and H-3 (*δ*_H_ 3.20) with H_3_-12 indicated H_3_-12 was *α*-oriented. The 11.2 Hz coupling constant between H-8′ and H-9′ and the NOE correlation of H-9′ with H_3_-11′ suggested the assignment of *α*-oriention for H-8′ and β-oriention for H-9′. The NOE correlation of OH-4′ with H-8′ indicated their cofacial *α*-orientations. The NOE correlations of H-3′ (*δ*_H_ 2.94) with H-8′ and H_3_-12′ indicated H_3_-12′ was also *α*-oriented. The NOE correlations of H-8 with OH-4′ indicated that H-8 and OH-4′were *α*-orientated. The absolute configuration of **4** was further determined by ECD calculation. The great similarity between the experimental ECD spectrum of **4** and the calculated ECD spectrum of (2*R*, 8*S*, 9*S*, 2′*R*, 4′*S*, 8′*R*, 9′*R*)-**4** (Figure 4) indicated the absolute configuration of **4** as shown.

Lecanicillium E (**5**) was obtained as a white powder with the molecular formula C_24_H_26_N_2_O_8_S_3_ as determined by HRESIMS ([M + NH_4_]^+^
*m*/*z* 584.1189). The ^1^H and ^13^C NMR spectra of **5** (Table 1 and Table 2) showed great similarity to those of bisdethiobis(methylthio)acetylaranotin [8]. The only obvious difference between them was the down-field shifts of H_3_-12′ (*δ*_H_ 2.47, ^1^H, s), C-12′ (*δ*_C_ 24.1) and C-2′ (*δ*_C_ 74.9) in **5**. Combined with the molecular formula, these data indicated a disulfide methyl attached at C-2′ instead of a sulfide methyl in **5**. Combined with the coupling constants of *J*_H-8′-H-9′_ and *J*_H-8-H-9_ (both 8.2 Hz), the NOE correlations of H-8 with H_3_-12, H-9 with H_3_-11, H-8′ with H_3_-12′, and H-9′ with H_3_-11′ indicated that H-8, H-8′, H_3_-12′, H_3_-12 were *α*-oriented, and H-9, H-9′, H_3_-11′, H_3_-11 were β-oriented (Appendix A). The absolute configuration of **5** was determined as shown by comparing the experimental ECD spectrum of **5** and the calculated ECD spectrum of (2*R*, 8*S*, 9*S*, 2′*R*, 8′*S*, 9′*S*)-**5** (Figure 4), which was further supported by the great similarity of the ECD spectra of **5** with bisdethiobis(methylthio)acetylaranotin [24].

Lecanicillium F (**6**) was isolated as a white solid with the molecular formula C_19_H_20_N_2_O_2_S as determined by HRESIMS ([M + H]^+^
*m*/*z* 341.1316). The ^1^H and ^13^C NMR spectra of **6** (Table 1 and Table 2) exhibited great similarity to those of emethacin B [17]. The only obvious difference between them was the disappearance of a sulfide methyl and the additional presence of a methine (*δ*_C_ 54.8, *δ*_H_ 3.26) instead of a nonprotonated carbon in **6**. The COSY correlations of *δ*_H_ 3.26 with H-4 and NH-1′ (*δ*_H_ 8.23), and the HMBC correlations from *δ*_H_ 3.26 to C-2 (*δ*_C_ 166.2), C-4 (*δ*_C_ 37.2), C-5 (*δ*_C_ 135.9), from NH-1 (*δ*_H_ 8.66) to C-2′ (*δ*_C_ 164.8), C-3′ (*δ*_C_ 68.2), C-4′ (*δ*_C_ 44.4), and from NH-1′ to C-2, C-3′ (Appendix A), suggested the methine was C-3 (*δ*_C_ 54.8, *δ*_H_ 3.26) in **6**. The NOE correlations of H-3 with H_3_-11′, and H-4 (*δ*_H_ 2.70) with H-4′ (*δ*_H_ 3.28) (Appendix A) indicated the cofacial orientations for H-3 and H_3_-11′, and H-4 (*δ*_H_ 2.70) and H-4′ (*δ*_H_ 3.28), respectively. The similarity between the experimental ECD spectrum of **6** and the calculated ECD spectrum of (3*S*,3′*R*)-**6** (Figure 4) indicated the absolute configuration of **6** as shown.

Compounds **2**–**19** were tested for their cytotoxicity towards human lung adenocarcinoma cell line H1975 and human hepatocellular carcinoma cell line HepG-2, and toxicity towards brine shrimps. The cytotoxicity results (Table 3) showed that **5**, **7**, and **16** displayed significant cytotoxicity against H1975, with IC_50_ values of 7.2~16.9 μM, **4**, **11**, **17**, and **18** displayed mild cytotoxicity against H1975, with IC_50_ values of 35.2~71.5 μM; only **18** had mild cytotoxicity against HepG-2, with an IC_50_ value of 41.2 μM. The results indicated that the disulfide bond unit in **5** was an active group for its cytotoxicity, which is consistent with the conclusions reported in the literatures [25,26]. A comparison of the structures and cytotoxicities of **6**, **7**, and **19** suggested that lacking thiomethyl and benzene could significantly decrease the cytotoxicity of this type of alkaloids. In addition, a comparison of the structures and cytotoxicity of **2**, **3**, **4**, **11**, and **16** suggests the skeleton of A/B/C ring fragment also could significantly affect the cytotoxicity of this type of alkaloids. Furthermore, brine shrimp lethality assays (Table 3) showed that only **4** and **15** exhibited medium toxicity, with TC_50_ values of 40–50 μM, towards brine shrimps. 

Compounds **1**–**19** were also evaluated for their antibacterial activity towards eight pathogens: *Bacillus subtilis*, *Micrococcus luteus*, *Escherichia coli*, *Staphylococcus aureus*, *S. aureus* MRSA, *Streptococcus agalactiae*, *S. iniae*, and *Pseudomonas aeruginosa*. Antibacterial assays (Table 3) exhibited that only **5** had significant antibacterial activity against *B. subtilis*, *M. luteus*, *S. agalactiae*, and *S. iniae*, with MIC values of 10~40 μg/mL. Other compounds showed mild or no obvious antibacterial activity. The results indicated that a disulfide bond unit at C-2′ was crucial for the antibacterial activity of this type of alkaloids.

## 3. Experimental Section

### 3.1. General Experimental Procedure

UV spectra were measured using a UV-2600 spectrophotometer (Shimadzu). IR spectra were obtained on an IR Affinity-1 Fourier transform infrared spectrophotometer (Shimadzu, Kyoto, Japan). ECD spectra were acquired on a Chirascan circular dichroism spectrometer (Applied Photophysics Ltd., Graz, Austria). Optical rotations were recorded using a MCP 500 polarimeter (Anton Paar). Melting points were recorded with a digital display microscopic melting point instrument (SGW X-5). NMR data were acquired with a Bruker AVANCE III HD 700 MHz NMR spectrometer (Bruker) with TMS as reference. HRESIMS spectroscopic data were obtained on a MaXis quadrupole-time-of-flight mass spectrometer (Bruker, Karlsruhe, Germany). Preparative reversed-phase HPLC was performed on a Shimadzu LC-20A preparative liquid chromatography system with a YMC-Pack ODS column (250 × 20 mm, S-5 μm, 12 nm). Sephadex LH-20 (GE Healthcare) was used for the chromatographic column (CC). RP-MPLC (reversed-phase-medium pressure preparative liquid chromatography) was carried out using the CHEETAH MP200 system (Agela Technologies, Tianjin, China) and Claricep Flash columns filled with ODS (40-63 μm, YMC). Silica gel (200–300 mesh) for CC and GF254 for TLC were purchased from Yantai Jiangyou Silica Gel Development Co., Ltd. Sea salts were commercially obtained from Guangzhou Hai Li Aquarium Technology Co., Ltd., Guangzhou, China.

### 3.2. Fungal Material

The fungus *Lecanicillium kalimantanense* was isolated from a mangrove sediment sample collected in the Bailu park, Sanya city, Hainan province. The strain was identified as *Lecanicillium kalimantanense* by internally transcribed spacer (ITS) region sequence data of the rDNA and given the Genbank accession number KM264285. The fungus *L. kalimantanense* was deposited in the RNAM Center, South China Sea Institute of Oceanology, Chinese Academy of Science.

### 3.3. Fermentation and Extraction

The spores of the fungus *L. kalimantanense* were added to 5 × 500 mL Erlenmeyer flasks, each containing 200 mL potato dextrose (PD) medium, and fermented for 3 days at 28 °C. Then 3 mL of spore suspension was transferred into 267 × 1 L Erlenmeyer flasks, each containing 300 mL culture media (glucose 1%, D-mannitol 2%, maltose 2%, corn meal 0.05%, monosodium glutamate 1%, KH_2_PO_4_ 0.05%, MgSO_4_·7H_2_O 0.03%, yeast extract 0.3%, sea salt 3%). Static fermentation was performed for 26 days at 28 °C. After fermentation, the broth and mycelia were separated with gauze. The broth was extracted with EtOAc to obtain crude extract (28.9 g). The mycelia were extracted three times with acetone, and further extracted three times with EtOAc to yield a crude extract (58.3 g). Then the two crude extracts (28.9 g and 58.3 g) were combined for further isolation.

### 3.4. Isolation and Purification

The combined crude extract (87.2 g) was fractionated on a normal-phase column using a stepped gradient elution with CH_2_Cl_2_/MeOH (*v*/*v*, 100:0, 98:2, 95:5, 90:10, 85:15, 80:20, 70:30, 50:50, 0:100) to obtain eight fractions (Fr.1–Fr.8). Fr.3 (10.9 g) was separated with Sephadex LH-20 eluting with CH_2_Cl_2_/MeOH (1:1) to obtain eleven subfractions (Fr.3.1–Fr.3.11). Fr.3.3 was further purified by HPLC eluting with MeOH/H_2_O (*v*/*v* 7:3, 3 mL/min) to give **12** (9.0 mg, t_R_ = 13.0 min). Fr.3.4 was further purified by HPLC eluting with MeOH/H_2_O (*v*/*v* 6:4, 3 mL/min) to give **8** (3.5 mg, t_R_ = 29.0 min), **10** (11.0 mg, t_R_ = 22.0 min), and **13** (9.0 mg, t_R_ = 19.0 min). Fr.3.5 was further purified by HPLC eluting with MeOH/H_2_O (*v*/*v* 55:45, 3 mL/min) to give **5** (1.7 mg, t_R_ = 43.0 min), **7** (1.1 mg, t_R_ = 35.5 min), and **15** (2.6 mg, t_R_ = 44.0 min). Fr.3.9 was further purified by HPLC eluting with MeOH/H_2_O (*v*/*v* 7:3, 3 mL/min) to give **9** (20.0 mg, t_R_ = 10.0 min). Then Fr.3.6–Fr.3.8 were combined and further separated by ODS column eluting with MeOH/H_2_O (*v*/*v* 10:90–100:0) to obtain subfractions (Fr.3.6.1–Fr.3.6.22). Fr.3.6.6 was purified by HPLC eluting with MeOH/H_2_O (*v*/*v* 42.5:57.5, 2.5 mL/min) to give **6** (2.3 mg, t_R_ = 32.0 min). Fr.3.6.8 was purified by HPLC eluting with MeOH/H_2_O (*v*/*v* 55:45, 2.5 mL/min) to give **3** (1.6 mg, t_R_ = 26.0 min), **2** (1.5 mg, t_R_ = 30.0 min), and **4** (2.6 mg, t_R_ = 24.0 min). Fr.3.2.10 was purified by HPLC eluting with MeOH/H_2_O (*v*/*v* 58.5:41.5, 3 mL/min) to give **14** (3.0 mg, t_R_ = 33.0 min). Fr.3.6.12 was purified by HPLC eluting with MeOH/H_2_O (*v*/*v* 59:41, 3 mL/min) to give **11** (6.5 mg, t_R_ = 40.0 min) and **18** (3.4 mg, t_R_ = 45.0 min). Fr.3.6.15 was further purified by HPLC eluting with MeOH/H_2_O (*v*/*v* 6:4, 2.5 mL/min) to give **16** (0.8 mg, t_R_ = 75.0 min) and **17** (1.4 mg, t_R_ = 68.0 min). Fr.4 (8.6 g) was separated with Sephadex LH-20 eluting with CH_2_Cl_2_/MeOH (1:1) to obtain eleven subfractions (Fr.4.1–Fr.4.3). Fr.4.2 was further separated by ODS column eluting with MeOH/H_2_O (*v*/*v* 10:90–100:0) to obtain subfractions (Fr.4.2.1–Fr.4.2.33). Fr.4.2.8 was purified by HPLC eluting with MeOH/H_2_O (*v*/*v* 9:11, 3 mL/min) to give **19** (2.0 mg, t_R_ = 20.0 min). Fr.4.2.10 was purified by HPLC eluting with MeOH/H_2_O (*v*/*v* 9:11, 3 mL/min) to give **1** (1.5 mg, t_R_ = 26.0 min).

Lecanicillium A (**1**). White powder; [*α*${]}_{D}^{25}$ −86 (*c* 0.10, MeOH); UV (MeOH) *λ*_max_ (log *ε*) 212 (4.30), 277 (3.70), 283 (3.60) nm; ECD (0.33 mM, MeOH) *λ*_max_ (∆*ε*) 200 (−9.00), 223 (16.86), 258 (−6.02) nm; IR (film) *ν*_max_ 3746, 3433, 3267, 2922, 2855, 2365, 2322, 2257, 1676, 1458, 1389, 1236, 1190, 1109, 1084, 1022, 995, 862, 760, 644, 596 cm^−1^; ^1^H and ^13^C NMR data, Table 1 and Table 2; HR-ESIMS *m*/*z* 373.0851 [M + H]^+^ (calcd for C_18_H_17_N_2_O_5_S, 373.0853).

Lecanicillium B (**2**). White powder; [*α*${]}_{D}^{25}$ −168 (*c* 0.10, MeOH); UV (MeOH) *λ*_max_ (log *ε*) 210 (4.30), 276 (3.53), 283 (3.47) nm; ECD (0.32 mM, MeOH) *λ*_max_ (∆*ε*) 200 (21.64), 201 (34.30), 229 (−34.46) nm; IR (film) *ν*_max_ 3341, 2922, 2841, 1676, 1649, 1545, 1514, 1460, 1410, 1113, 1018, 671, 598 cm^−1^; ^1^H and ^13^C NMR data, Table 1 and Table 2; HR-ESIMS *m*/*z* 387.1009 [M + H]^+^ (calcd for C_19_H_19_N_2_O_5_S, 387.1009).

Lecanicillium C (**3**). colorless crystals; mp 172–174 °C; [*α*${]}_{D}^{25}$ −213 (*c* 0.10, CH_3_CN); UV (CH_3_CN) *λ*_max_ (log *ε*) 206 (4.50) nm; ECD (0.45 mM, CH_3_CN) *λ*_max_ (∆*ε*) 200 (−24.56), 201 (−17.74), 203 (−25.16), 214 (−11.98), 227 (−31.84), 247 (−3.68), 257 (−6.35) nm; IR (film) *ν*_max_ 3861, 3744, 3618, 2922, 2853, 1738, 1674, 1514, 1377, 1236, 1194, 1138, 1034, 972, 731, 648, 602 cm^−1^; ^1^H and ^13^C NMR data, Table 1 and Table 2; HR-ESIMS *m*/*z* 573.0969 [M + Na]^+^ (calcd for C_24_H_26_N_2_NaO_9_S_2_, 573.0972).

Lecanicillium D (**4**). White powder; [*α*${]}_{D}^{25}$ −206 (*c* 0.10, MeOH); UV (MeOH) *λ*_max_ (log *ε*) 207 (4.35) nm; ECD (0.45 mM, MeOH) *λ*_max_ (∆*ε*) 200 (3.58), 226 (−33.79), 247 (−8.64), 256 (−10.55) nm; IR (film) *ν*_max_ 3331, 2945, 2920, 2841, 1740, 1659, 1535, 1514, 1460, 1398, 1113, 1018, 671, 598 cm^−1^; ^1^H and ^13^C NMR data, Table 1 and Table 2; HR-ESIMS *m*/*z* 568.1408 [M + NH_4_]^+^ (calcd for C_24_H_30_N_3_O_9_S_2_, 568.1418).

Lecanicillium E (**5**). White powder; [*α*${]}_{D}^{25}$ −282 (*c* 0.10, MeOH); UV (MeOH) *λ*_max_ (log *ε*) 207 (4.36) nm; ECD (0.44 mM, MeOH) *λ*_max_ (∆*ε*) 200 (9.37), 202 (−2.30), 225 (−61.07), 247 (−17.03), 256 (−19.43) nm; IR (film) *ν*_max_ 3347, 2951, 2924, 2843, 1734, 1670, 1375, 1302, 1236, 1192, 1130, 1018, 669, 598 cm^−1^; ^1^H and ^13^C NMR data, Table 1 and Table 2; HR-ESIMS *m*/*z* 584.1189 [M + NH_4_]^+^ (calcd for C_24_H_30_N_3_O_8_S_3_, 584.1190).

Lecanicillium G (**6**). White powder; [*α*${]}_{D}^{25}$ −5 (*c* 0.10, MeOH); UV (MeOH) *λ*_max_ (log *ε*) 206 (4.08) nm; ECD (0.44 mM, MeOH) *λ*_max_ (∆*ε*) 200 (−19.33), 201 (16.40), 204 (21.49), 228 (−10.86) nm; IR (film) *ν*_max_ 3354, 2943, 2833, 1670, 1653, 1558, 1541, 1506, 1472, 1456, 1418, 1115, 1020, 667, 599 cm^−1^; ^1^H and ^13^C NMR data, Table 1 and Table 2; HR-ESIMS *m*/*z* 341.1316 [M + H]^+^ (calcd for C_19_H_21_N_2_O_2_S, 341.1318).

### 3.5. X-ray Crystallographic Analysis of ***3***

Crystallographic data were collected on a Rigaku MicroMax 007 diffractometer (Rigaku Corporation, Tokyo, Japan) equipped with Cu Kα radiation and a graphite monochromator. The structure was solved by direct methods with the SHELXTL and refined by full-matrix, least-squares techniques. Crystallographic data for **3** were deposited with the Cambridge Crystallographic Data Centre as supplementary publication number CCDC 2298244.

Crystal data for **3**: C_24_H_26_N_2_O_9_S_2_·CH_3_OH, *FW* = 582.63; colorless crystal from MeOH; crystal size = 0.15 × 0.12 × 0.1 mm^3^; *T* = 100.00 (10) K; monoclinic, space group P2_1_ (no. 4); unit cell parameters: *a* = 7.14328(5) Å, *b* = 16.25526(10) Å, *c* = 11.41675(7) Å, *α* = 90°, β = 93.4877(6)°, *γ* = 90°, *V* = 1323.211(14) Å^3^, *Z* = 2, *D*_calc_ = 1.462 g/cm^3^, *F* (000) = 612.0, μ (CuK*α*)= 2.357 mm^−1^; 25678 reflections measured (7.758° ≤ 2θ ≤ 148.752°), 5292 unique (*R*_int_ = 0.0323, R_sigma_ = 0.0216), which were used in all calculations. The final R_1_ was 0.0242 (I > 2σ(I)) and *w*R_2_ was 0.0641 (all data). Flack parameter = −0.001(4).

### 3.6. ECD Calculations

The ECD calculations for **1**–**6** were performed using the Gaussian 16 program package. The procedures were the same as described in our previous studies [27]. The Spartan 14 program (Wavefunction Inc., Tokyo, Japan) was used for calculating the molecular Merck force field (MMFF). Density functional theory (DFT) and time-dependent density functional theory (TDDFT) calculations were performed using the Gaussian 16 program package. An MMFF model was used for the conformational search, then the conformers with lower relative energies (<10 kcal/mol) were subjected to geometry optimization with the DFT method at the B3LYP/6-31g (d) level in MeOH or CH_3_CN. Vibrational frequency calculations were carried out at the same level to evaluate their relative thermal (ΔE) and free energies (ΔG) at 298.15 K. The geometry optimized conformers were further calculated at the M062X/def2TZVP level and the solvent (MeOH or CH_3_CN) effects were taken into consideration by using SMD. The optimized conformers with a Boltzmann distribution of more than 1% population were further subjected to ECD calculation, which were performed by TDDFT methodology at the PBE1PBE/6-311g(d) level. The number of excited states was 60 for **1**, **2**, and **6**, 85 for **3** and **4**, and 102 for **5**. The ECD spectra were generated by the software SpecDis using a Gaussian band shape with 0.20–0.30 eV exponential half-width from dipole-length dipolar and rotational strengths. The equilibrium population of each conformer at 298.15 K was calculated from its relative free energies using Boltzmann statistics. The calculated spectra of compounds **1**–**6** were generated from the low-energy conformers according to the Boltzmann weighting of each conformer in MeOH solution for **1**, **2**, **4**, **5**, and **6** and CH_3_CN for **3**, respectively.

### 3.7. Cytotoxicity

Cytotoxic activity was evaluated using the human lung adenocarcinoma cell line H1975 and human hepatocellular carcinoma cell line HepG-2, using the CCK-8 method as described in our previous study [28]. Briefly, each of the test compounds was dissolved in DMSO and further diluted to give final concentrations of 80, 40, 20, 10, 5, 2.5, and 1.25 μM, respectively. H1975 cells or HepG-2 cells (5 *×* 10^3^ cells/plate) were seeded in 96-well plates and treated with compounds at the indicated concentration for 24 hours and then 10 μL CCK-8 reagent was added to each well. The plates were incubated at 37 °C for another 4 hours. Finally, the optical density was measured at a wavelength of 450 nm with a Bio-Rad microplate reader. Dose–response curves were generated, and the IC_50_ values were calculated from the linear portion of log dose–response curves.

### 3.8. Brine Shrimp Lethality Bioassays

The methods were the same as those used in our previous studies [27].

### 3.9. Antibacterial Assays

The micro broth dilution method [29] was used to evaluate the antibacterial activities of compounds **1**–**19** against the growth of eight common pathogens: *Bacillus subtilis* BS01, *Micrococcus luteus, Escherichia coli* ATCC 25922, *Staphylococcus aureus* ATCC 6538, *S. aureus* MRSA, *Streptococcus agalactiae* ATCC 13813, *S. iniae*, and *Pseudomonas aeruginosa* ATCC 9027 in 96-well polystyrene plates. Vancomycin and ciprofloxacin were used as positive controls. Briefly, wells containing 100 μL bacteria dilutions (OD_600_ = 0.01) in Luria–Bertani (LB) medium were supplemented with different concentrations of **1**–**19** (80, 40, 20, 10, 5, 2.5, 1.25, and 0.625 μg/mL), vancomycin, and ciprofloxacin (40, 20, 10, 5, 2.5, 1.25, 0.625, and 0.3125 μg/mL), respectively. The MICs were determined after 24 hours’ incubation at 37 °C with the tested compounds.

## 4. Conclusions

In summary, six new thiodiketopiperazine-class alkaloids, lecanicilliums A–F (**1**–**6**), together with 13 known analogues (**7**–**19**) were isolated from the mangrove sediment-derived fungus *L. kalimantanense* SCSIO41702. Lecanicillium A contained an unprecedented 6/5/6/5/7/6 cyclic system with a spirocyclic center at C-2′. Biologically, lecanicillium E, emethacin B, and versicolor A displayed significant inhibitory activity against H1975, with IC_50_ values of 7.2~16.9 μM, and lecanicillium E also showed antibacterial activities against four pathogens, with MIC values of 10~40 μg/mL. The cytotoxicity and antibacterial activity results indicated that the disulfide bond unit at C-2′ was crucial for the activity of this kind of alkaloids. This finding further clarified the chemical structure diversity of thiodiketopiperazine-class alkaloids, and the structural diversity and biological activities of thiodiketopiperazine-class alkaloids may be worthy of further studies.

## Figures and Tables

**Figure 1 marinedrugs-21-00575-f001:**
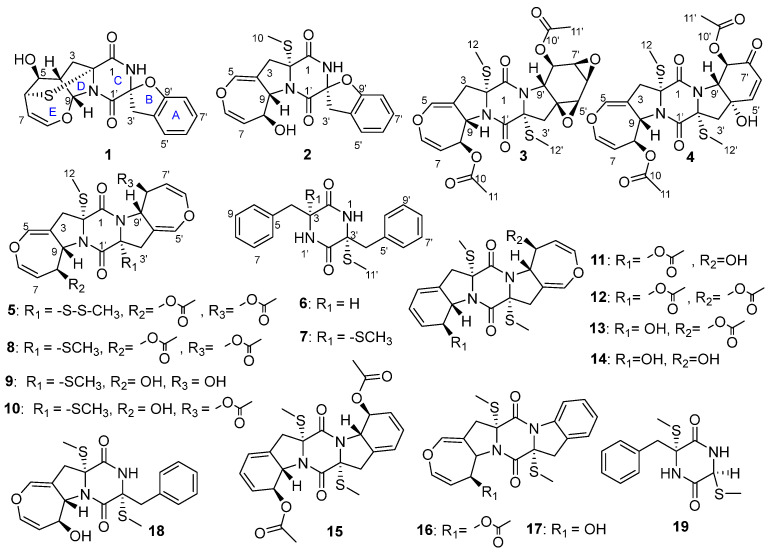
Structures of compounds **1**–**19** isolated from *L. kalimantanense* SCSIO41702.

**Figure 2 marinedrugs-21-00575-f002:**
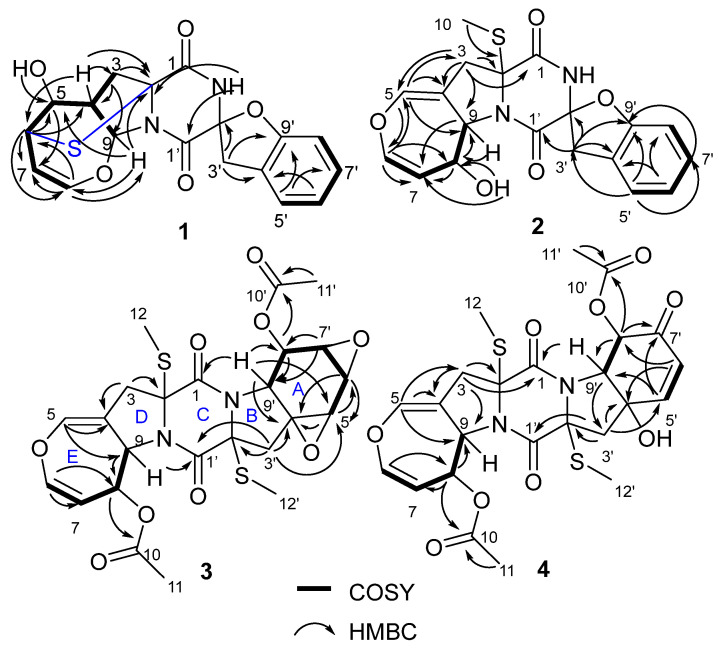
Key HMBC and COSY correlations of compounds **1**–**4.**

**Figure 3 marinedrugs-21-00575-f003:**
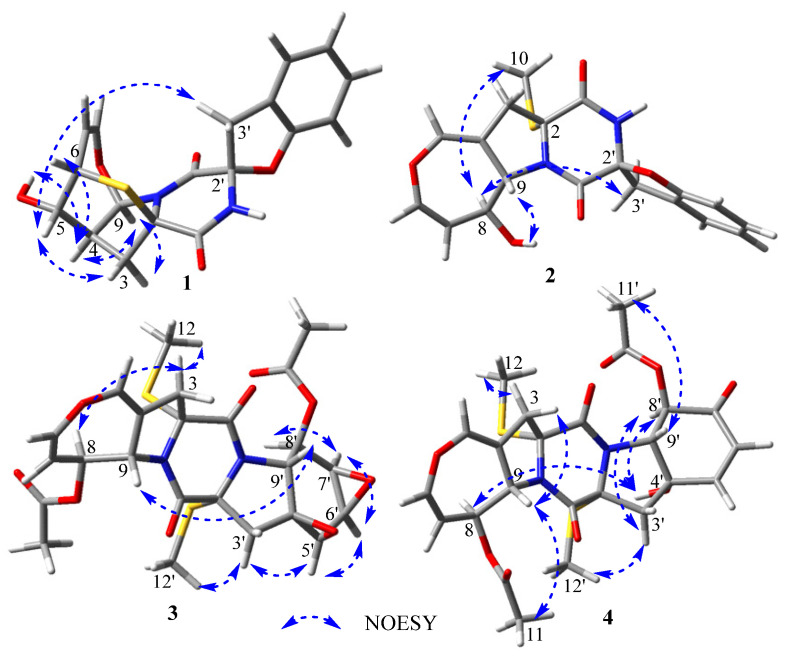
Key NOESY correlations of compounds **1**–**4**.

**Figure 4 marinedrugs-21-00575-f004:**
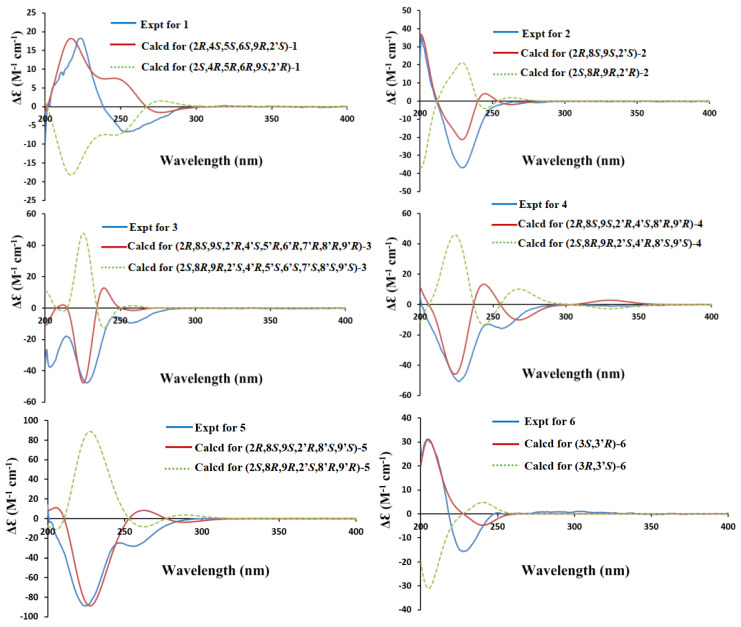
Comparison of experimental and calculated ECD spectra of **1**, **2**, **4**–**6** in MeOH and **3** in CH_3_CN.

**Figure 5 marinedrugs-21-00575-f005:**
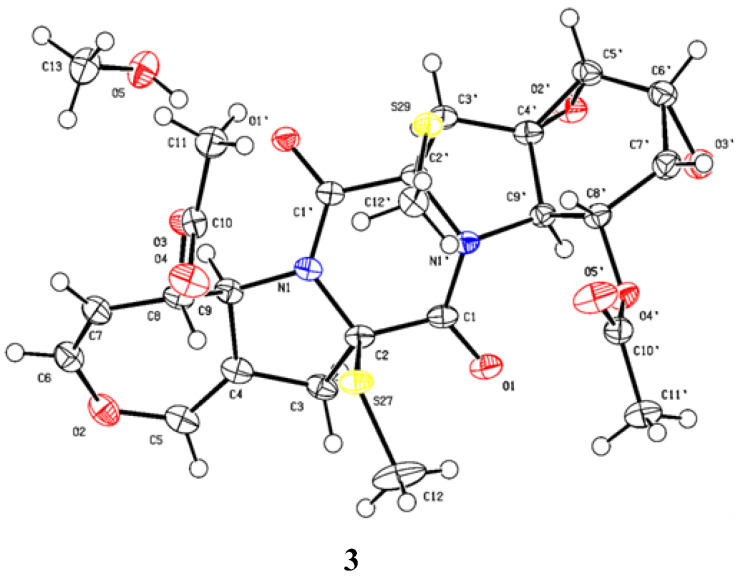
ORTEP plot of the X-ray crystallographic data for **3**.

**Table 1 marinedrugs-21-00575-t001:** ^1^H NMR data of compounds **1**–**6** (700 MHz, *δ*_H_ in ppm, *J* in Hz) in DMSO-*d*_6._

No.	1	2	3	4	5	6
3	2.21 (d, 12.7), 2.71 (dd, 12.7, 6.5)	2.91 (dt, 15.3, 1.2), 3.14 (dt, 15.3, 2.2)	2.96 (dt, 15.2, 1.2) 3.25 (dt, 15.2, 2.3)	2.92 (dd, 16.1, 2.1) 3.20 (d, 16.1)	2.96 (dt, 15.4, 1.2) 3.24 (dt, 15.4, 2.3)	3.26 (overlap, m)
4	3.06 (t, 6.5)	-	-	-	-	2.70 (dd, 13.6, 5.2), 3.05 (dd, 13.6, 4.5)
5	3.87 (d, 6.5)	6.69 (t, 2.4)	6.76 (t, 2.6)	6.81 (t, 2.6)	6.77 (t, 2.2)	-
6	4.86 (ddd, 7.8, 6.5, 1.2)	6.29 (dd, 8.2, 2.4)	6.43 (dd, 8.3, 2.6)	6.43 (dd, 8.2, 2.6)	6.42 (dd, 8.2, 2.0)	7.19 (overlap, m)
7	4.82 (ddd, 7.8, 6.5, 1.2)	4.79 (dd, 8.2, 2.1)	4.70 (dd, 8.3, 2.2)	4.69 (dd, 8.2, 2.2)	4.72 (dd, 8.2, 2.0)	7.26 (overlap, m)
8	6.00 (d, 6.5)	4.46 (ddt, 7.7, 6.7, 2.1)	5.57 (dt, 8.1, 2.2)	5.52 (dt, 8.2, 2.2)	5.54 (dt, 8.2, 2.1)	7.18 (overlap, m)
9	6.61 (d, 6.5)	4.74 (dq, 7.7, 2.1)	4.92 (dq, 8.1, 2.2)	5.00 (dq, 8.2, 2.2)	4.99 (dq, 8.2, 2.1)	7.26 (overlap, m)
10	-	2.29 (s)	-	-	-	7.19 (overlap, m)
11	-	-	2.01 (s)	1.97 (s)	1.95 (s)	-
12	-	-	2.13 (s)	2.18 (s)	2.15 (s)	-
3′	3.26 (d, 16.4), 3.94 (d, 16.4)	3.25 (d, 16.6), 4.01 (d, 16.6)	2.43 (d, 14.4) 2.85 (d, 14.4)	2.94 (d, 15.4) 3.16 (dd, 15.4, 2.1)	3.30 (overlap, m) 3.45 (dt, 15.9, 1.3)	-
4′	-	-	-	-	-	2.84 (d, 13.1), 3.28 (d, 13.1)
5′	7.26 (d, 7.5)	7.26 (d, 7.5)	3.83 (d, 2.7)	6.98 (d, 10.3)	6.86 (t, 2.2)	-
6′	6.92 (t, 7.5)	6.91 (t, 7.5)	3.67 (dd, 4.3, 2.7)	6.09 (d, 10.3)	6.43 (dd, 8.2, 2.0)	7.12 (overlap, m)
7′	7.14 (t, 8.0)	7.14 (t, 8.0)	3.34 (dd, 4.3, 1.2)	-	4.68 (dd, 8.2, 2.0)	7.24 (overlap, m)
8′	6.81 (d, 8.0)	6.79 (d, 8.0)	5.79 (dd, 9.1, 1.2)	5.73 (d, 11.2)	5.52 (dt, 8.2, 2.2)	7.26 (overlap, m)
9′	-	-	4.34 (d, 9.1)	4.92 (dd, 11.2, 1.5)	4.95 (dq, 8.2, 2.2)	7.24 (overlap, m)
10′	-	-	-	-	-	7.12 (overlap, m)
11′	-	-	1.96 (s)	2.07 (s)	1.97 (s)	1.37 (s)
12′	-	-	2.25 (s)	2.16 (s)	2.47 (s)	-
NH	10.06 (s)	9.77 (s)	-	-	-	-
1-NH	-	-	-	-	-	8.66 (s)
1′-NH	-	-	-	-	-	8.23 (d, 2.0)
5-OH	5.31 (s)	-	-	-	-	-
8-OH	-	5.28 (d, 6.7)	-	-	-	-
4′-OH	-	-	-	6.18 (s)	-	-

**Table 2 marinedrugs-21-00575-t002:** ^13^C NMR data of compounds **1**–**6** (175 MHz, *δ*_C_ in ppm) in DMSO-*d*_6_.

No.	1	2	3	4	5	6
1	165.2, C	166.4, C	163.5, C	164.3, C	163.7, C	-
2	70.8, C	69.2, C	70.1, C	70.8, C	70.7, C	166.2, C
3	48.2, CH_2_	40.2, CH_2_	38.7, CH_2_	40.0, CH_2_	39.0, CH_2_	54.8, C
4	46.3, CH	109.1, C	110.7, C	110.4, C	110.9, C	37.2, CH_2_
5	69.9, CH	136.7, CH	137.0, CH	137.6, CH	137.1, CH	135.9, C
6	52.5, CH	137.9, CH	139.9, CH	139.8, CH	139.9, CH	130.5, CH
7	100.6, CH	111.0, CH	105.3, CH	105.8, CH	105.5, CH	128.0, CH
8	144.9, CH	71.2, CH	71.2, CH	71.5, CH	71.2, CH	126.6, CH
9	92.9, CH	64.1, CH	59.8, CH	59.6, CH	59.6, CH	128.0, CH
10	-	13.3, CH_3_	169.6, C	169.4, C	169.4, C	130.5, CH
11	-	-	20.7, CH_3_	20.8, CH_3_	20.9, CH_3_	-
12	-	-	13.8, CH_3_	14.1, CH_3_	13.9, CH_3_	-
1′	162.8, C	163.4, C	163.6, C	163.1, C	164.3, C	-
2′	91.8, C	93.0, C	71.7, C	69.2, C	74.9, C	164.8, C
3′	38.3, CH_2_	38.4, CH_2_	39.0, CH_2_	48.0, CH_2_	37.6, CH_2_	68.2, C
4′	125.4, C	125.5, C	60.9, C	75.5, C	110.6, C	44.4, CH_2_
5′	124.5, CH	124.5, C	53.6, CH	150.2, CH	137.3, CH	135.0, C
6′	121.3, CH	121.2, CH	47.9, CH	125.8, CH	139.9, CH	130.3, CH
7′	128.1, CH	128.0, CH	54.1, CH	191.5, C	105.5, CH	128.0, CH
8′	109.1, CH	109.1, CH	72.6, CH	74.8, CH	70.7, CH	127.1, CH
9′	156.6, C	156.7, CH	57.5, CH	69.1, CH	60.1, CH	128.0, CH
10′	-	-	169.4, C	168.9, C	169.5, C	130.3, CH
11′	-	-	20.6, CH_3_	20.3, CH_3_	20.7, CH_3_	11.2, C
12′	-	-	14.3, CH_3_	13.8, CH_3_	24.1, CH_3_	-

**Table 3 marinedrugs-21-00575-t003:** Antibacterial activities, cytotoxicity, and toxicity of **1**–**19**.

Comp.	Antibacterial Activity (MIC: μg/mL)								Cytotoxicity (IC_50_)	against (μM)	Toxicity (TC_50_ in μg/mL)
	*B. subtilis*	*M.* *luteus*	*E.* *coli*	*S.* *aureus*	MRSA	*S. agalactiae*	*P. aeruginosa*	*S.* *iniae*	H1975	HepG-2	Brine Shrimp
**1**	60	>80	>80	>80	>80	>80	>80	>80	-	-	>80
**2**	80	60	>80	>80	>80	>80	>80	70	>80	>80	>80
**3**	>80	>80	>80	>80	>80	>80	>80	>80	>80	>80	>80
**4**	50	>80	>80	>80	>80	>80	>80	40	71.5	>80	80
**5**	10	40	80	>80	>80	25	>80	15	16.3	>80	40
**6**	>80	>80	>80	>80	>80	>80	>80	>80	>80	>80	-
**7**	>80	>80	>80	>80	>80	>80	>80	>80	16.9	>80	-
**8**	>80	>80	>80	>80	>80	>80	>80	>80	>80	>80	-
**9**	>80	>80	>80	>80	>80	>80	>80	>80	>80	>80	>80
**10**	>80	>80	>80	>80	>80	>80	>80	>80	>80	>80	>80
**11**	60	>80	>80	>80	>80	>80	>80	50	69.3	>80	>80
**12**	>80	50	>80	>80	>80	50	>80	>80	>80	>80	50
**13**	>80	>80	>80	>80	>80	>80	>80	>80	>80	>80	-
**14**	>80	>80	>80	>80	>80	>80	>80	40	>80	>80	>80
**15**	80	>80	>80	>80	>80	>80	>80	>80	>80	>80	>80
**16**	70	>80	>80	>80	>80	70	>80	>80	7.2	>80	-
**17**	80	>80	>80	>80	>80	70	>80	30	35.2	>80	-
**18**	80	>80	>80	>80	>80	>80	>80	40	50.6	41.2	>80
**19**	80	70	>80	>80	>80	>80	>80	>80	>80	>80	-
VMN	-	0.6	-	-	2.5	-	-	-	-	-	-
CFN	0.6	-	0.6	0.6	-	5	0.6	2.5	-	-	-

VMN: Vancomycin; CFN: Ciprofloxacin; “-”: Not tested.

## Data Availability

The original data presented in the study are included in the article/Appendix A; further inquiries can be directed to the corresponding author.

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
