# Peer review of "Lecanicilliums A–F, Thiodiketopiperazine-Class Alkaloids from a Mangrove Sediment-Derived Fungus Lecanicillium kalimantanense"

_marinedrugs, 2023, doi:10.3390/md21110575_

Round 1

Reviewer 1 Report

Comments and Suggestions for Authors

1. Page 3, Line 3. “four tertiary methyl groups at δH 6.61” should be “four tertiary methine groups at δH 6.61”

2. Page 3, Line 4. The signal multiplicity was incorrectly indicated at 3.87 (1H, d, J = 6.5 Hz).

3. Page 3, Last line. “The coupling constant of JH-7-H-8 = 7.8 Hz indicated …” However, in Table 1, the coupling constant value was indicated as 6.5 Hz.

4. Page 4, second paragraph. “(CH), 144.9 (CH), and 92.9 (CH) in 2, respectively).” Incorrect chemical shift values are indicated.

5. Page 5. “presence of three tertiary methyl signals (δH 3.83 (1H, d, J = 2.7 Hz), 3.67 (1H, dd, J = 4.3, 2.7 Hz), 3.34 (1H, dd, J = 4.3, 1.2 Hz); δC 54.1, 53.6, 47.9)” There are no such methyl groups in the structure of compound 5.

6. Page 6. “and the disappearance of three tertiary methyl signals in 4.” The authors mixed up methyl and methine groups everywhere.

7. Page 6. “The NOE correlations of H-8 with H-8’ indicated that H-8 and H-8’were α-orientation.” A questionable NOE correlation is given. Because these protons are located far from each other.

8. Page 7. The NOE correlations of H-3 with H3-11’, and H2-4 with H2-4’ (Figure S65) indicated the cofacial orientations for H-3 and H3-11’, and H2-4 and H2-4’, respectively. Incorrect NOE correlation given for H2-4 and H2-4’. These methylene groups have two different protons.

9. Table 1. The coupling constant values (2.0 Hz) for olefin protons H-6 with H-7 and H-6` with proton H-7` in compound 5 are erroneously given.

Author Response

Responding to the comments of Reviewer 1

Q1. Page 3, Line 3. “four tertiary methyl groups at δH 6.61” should be “four tertiary methine groups at δH 6.61”

Answer: The error was revised.

Q2. Page 3, Line 4. The signal multiplicity was incorrectly indicated at 3.87 (1H, d, J = 6.5 Hz).

Answer: The signal multiplicity of H-5 (δH 3.87) was doublet. The expanded 1H NMR spectrum of 1 in 3.20-4.05 ppm was added in the supporting information.

Q3. Page 3, Last line. “The coupling constant of JH-7-H-8 = 7.8 Hz indicated …” However, in Table 1, the coupling constant value was indicated as 6.5 Hz.

Answer: The error was revised. “The coupling constant of JH-7-H-8 = 7.8 Hz indicated …” was revised to “The coupling constant of JH-7-H-8 = 6.5 Hz indicated …” in the revision.

Q4. Page 4, second paragraph. “(CH), 144.9 (CH), and 92.9 (CH) in 2, respectively).” Incorrect chemical shift values are indicated.

Answer: The error was revised. “144.9 (CH) and 92.9 (CH)” was revised to “71.2 (CH) and 64.1 (CH)” in the revision.

Q5. Page 5. “presence of three tertiary methyl signals (δH 3.83 (1H, d, J = 2.7 Hz), 3.67 (1H, dd, J = 4.3, 2.7 Hz), 3.34 (1H, dd, J = 4.3, 1.2 Hz); δC 54.1, 53.6, 47.9)” There are no such methyl groups in the structure of compound 5.

Answer: Page 5: These three tertiary methine signals are signals of compound 3. The error was revised.

Q6. Page 6. “and the disappearance of three tertiary methyl signals in 4.” The authors mixed up methyl and methine groups everywhere.

Answer: These error s were revised.

Q7. Page 6. “The NOE correlations of H-8 with H-8’ indicated that H-8 and H-8’were α-orientation.” A questionable NOE correlation is given. Because these protons are located far from each other.

Answer: “The NOE correlations of H-8 with H-8’ indicated that H-8 and H-8’were α-orientation.” was revised to “The NOE correlations of H-8 with OH-4’ indicated that H-8 and OH-4’were α-orientation.” in the revision. 

Q8. Page 7. The NOE correlations of H-3 with H3-11’, and H2-4 with H2-4’ (Figure S65) indicated the cofacial orientations for H-3 and H3-11’, and H2-4 and H2-4’, respectively. Incorrect NOE correlation given for H2-4 and H2-4’. These methylene groups have two different protons.

Answer: “The NOE correlations of H-3 with H3-11’, and H2-4 with H2-4’ (Figure S65) indicated the cofacial orientations for H-3 and H3-11’, and H2-4 and H2-4’, respectively.” was revised to “The NOE correlations of H-3 with H3-11’, and H-4 (δH 2.70) with H-4’ (δH 3.28) (Figure S65) indicated the cofacial orientations for H-3 and H3-11’, and H-4 (δH 2.70) and H-4’ (δH 3.28), respectively.” in the revision.

Q9. Table 1. The coupling constant values (2.0 Hz) for olefin protons H-6 with H-7 and H-6’ with proton H-7’ in compound 5 are erroneously given.

Answer: Table 1. “The coupling constant values (2.0 Hz) for olefin protons H-6 with H-7 and H-6’ with proton H-7’ in compound 5” was revised to be 8.2 Hz in the revision.

Reviewer 2 Report

Comments and Suggestions for Authors

The authors present 6 new compounds isolated from the fungus Lecanicillium kalimantanense SCSIO41702 together with thirteen known analogous. The structures were elucidated by NMR spectroscopy, electronic circular dichroism and single crystal X-ray diffraction. Some work was performed to find relevant biological activity for the compounds.

The work is suitable for publication in Marine Drugs after minor modifications:

1.      The authors claim “The absolute configuration of 3 was determined as 2R,8S,9S,2’R,4’S,5’R,6’R,7’R,8’R,9’R by a single crystal X-ray diffraction analysis”

The Flack parameter must be given in section 3.5 to support this statement. 

2.      Section 3.1. should be rewritten. The title and the short description are enigmatic.

Author Response

Responding to the comments of Reviewer 2

Q1. The authors claim “The absolute configuration of was determined as 2R,8S,9S,2’R,4’S,5’R,6’R,7’R,8’R,9’by a single crystal X-ray diffraction analysis”

The Flack parameter must be given in section 3.5 to support this statement.   Answer: “Flack parameter = -0.001(4).” was added in the revision.

Q2.  Section 3.1. should be rewritten. The title and the short description are enigmatic.
   Answer: Section 3.1. was rewritten. Please see the revision.

Reviewer 3 Report

Comments and Suggestions for Authors

Authors reported the structures of six new thiodiketopiperazine-class alkaloids from fungus and biological activities. The structural elucidation by NMR and ECD experiments are explained well. The activities are weak or moderate but the structures are very complicated. Those results are very important. Therefore, this manuscript is recommended to be published in Marine Drugs after minor revisions.

1)      The sulfide bond in the compound 1 was determined by chemical shifts of CH-6 and C-2. Wasn’t HMBC H6/C2 observed?

2)      Page 4, Line 17. There is a mistake about the chemical shifts of C-8 and C-9 in 2. “the chemical shifts of C-8 and C-9……. 144.9 (CH) and 92.9 (CH) in 2.” These chemical shifts are C-8 and C-9 in 1. 

3)      Figure 2. About HMBC of compound 3. It appears as if an arrow is drawn between C-2 and C-4.

4)      Page 7, Third line from the bottom. Use italic for scientific names of bacteria.

Author Response

Responding to the comments of Reviewer 3

Q1. The sulfide bond in the compound 1 was determined by chemical shifts of CH-6 and C-2. Wasn’t HMBC H6/C2 observed?

Answer: The HMBC correlation between H-6 and C-2  was observed, and this information was added in the revision.

Q2. Page 4, Line 17. There is a mistake about the chemical shifts of C-8 and C-9 in 2. “the chemical shifts of C-8 and C-9……. 144.9 (CH) and 92.9 (CH) in 2.” These chemical shifts are C-8 and C-9 in 1.

Answer: The error was revised, “144.9 (CH) and 92.9 (CH)” was revised to “71.2 (CH) and 64.1 (CH)” in the revision.

Q3. Figure 2. About HMBC of compound 3. It appears as if an arrow is drawn between C-2 and C-4.

Answer: The arrows from H-3 to C-4 and C-2 were drawn in the revision.

Q4. Page 7, Third line from the bottom. Use italic for scientific names of bacteria.

Answer: The error was revised.